# Analysis of the Spatiotemporal Changes of Ice Sheet Mass and Driving Factors in Greenland

**Yankai Bian [1,\*]** , **Jianping Yue [1]** , **Wei Gao [2]** , **Zhen Li [1]** , **Dekai Lu [1]** , **Yunfei Xiang [1]** and **Jian Chen [1]**

[1]  School of Earth Sciences and Engineering, Hohai University, Nanjing 211100, China; jpyue@hhu.edu.cn (J.Y.); lilzhenhhu@hhu.edu.cn (Z.L.); sdludekai@hhu.edu.cn (D.L.); yfxiang@hhu.edu.cn (Y.X.); 160211020002@hhu.edu.cn (J.C.)

[2]  School of Geology and Geomatics, Tianjin Chengjian University, Tianjin 300384, China; gaowei@tcu.edu.cn

\*  Correspondence: hhubyk.edu.cn@hhu.edu.cn; Tel.: +86-188-5200-0536

**Abstract:** With the warming of the global climate, the mass loss of the Greenland ice sheet is intensifying, having a profound impact on the rising of the global sea level. Here, we used Gravity Recovery and Climate Experiment (GRACE) RL06 data to retrieve the time series variations of ice sheet mass in Greenland from January 2003 to December 2015. Meanwhile, the spatial changes of ice sheet mass and its relationship with land surface temperature are studied by means of Theil–Sen median trend analysis, the Mann–Kendall (MK) test, empirical orthogonal function (EOF) analysis, and wavelet transform analysis. The results showed: (1) in terms of time, we found that the total mass of ice sheet decreases steadily at a speed of $-195 \pm 21$ Gt/yr and an acceleration of $-11 \pm 2$ Gt/yr$^2$ from 2003 to 2015. This mass loss was relatively stable in the two years after 2012, and then continued a decreasing trend; (2) in terms of space, the mass loss areas of the Greenland ice sheet mainly concentrates in the southeastern, southwestern, and northwestern regions, and the southeastern region mass losses have a maximum rate of more than 27 cm/yr (equivalent water height), while the northeastern region show a minimum rate of less than 3 cm/yr, showing significant changes as a whole. In addition, using spatial distribution and the time coefficients of the first two models obtained by EOF decomposition, ice sheet quality in the southeastern and northwestern regions of Greenland show different significant changes in different periods from 2003 to 2015, while the other regions showed relatively stable changes; (3) in terms of driving factors temperature, there is an anti-phase relationship between ice sheet mass change and land surface temperature by the mean XWT-based semblance value of $-0.34$ in a significant oscillation period variation of 12 months. Meanwhile, XWT-based semblance values have the largest relative change in 2005 and 2012, and the smallest relative change in 2009 and 2010, indicating that the influence of land surface temperature on ice sheet mass significantly varies in different years.

**Keywords:** Greenland ice sheet; GRACE; Theil–Sen median trend analysis; Empirical Orthogonal function analysis; wavelet transform analysis

## 1. Introduction

The Greenland ice sheet is the second largest ice sheet in the world after Antarctica, covering an area of 1.73 million square kilometers, and accounting for 10% of total global ice sheet volume [1]. In recent years, with the development of human civilization, the impact of human activities has become more pronounced on the mass balance of the Greenland ice sheet, which highlights serious environmental problems. Its melting rate is much faster than in the previous 20 years, contributing about 0.5 mm/yr to the current global sea level rise of about 3.2 mm/yr [2]. If this acceleration

continues, the total sea level rise caused by the melting of the Greenland ice sheet will reach about 9 cm by 2050 [3]. Therefore, the mass change of the Greenland ice sheet is of great significance to the study of global climate change, sea level change, and related fields; then, it is particularly important to understand the spatial–temporal distribution characteristics of ice sheet mass changes in Greenland, and especially its influencing factors.

In March 2002, the Gravity Recovery and Climate Experiment (GRACE) Satellite, launched by the Center for Space Research (CSR), made it possible to monitor the mass balance of the Greenland ice sheet. GRACE consists of two polar orbiting satellites. By detecting the distance between them, we can observe the spatial–temporal variation of the Earth's gravity field, and then obtain the redistribution of the Earth's surface mass [4–9]. In addition, the spatial resolution of the global gravity field is estimated to be several hundred kilometers and the temporal resolution is one month. At present, GRACE observational data have been well applied in many application fields, such as monitoring surface drought events [10–12], groundwater depletion [13–15], total water storage change [16,17], and glacier melting [18,19]. Among them, in terms of glacier melting, it is helpful to understand the impact on thermohaline circulation in the Atlantic Ocean. Murray found that the large freshwater input from the Greenland ice sheet may weaken or destroy the "thermo-salt" cycle of marine saltwater, thereby seriously altering the climate of the Northern Hemisphere [20]; Yang et al. estimated new heat and salt flows from the North Atlantic to the Labrador Sea using updated GRACE satellite data, and suggested that the changes in Labrador Sea Water Density (LSW) might be related to the weakening of the Atlantic Meridional Overturning Circulation (AMOC) [21]. In addition, it also plays an important role in the study of the mass change of the Greenland ice sheet.

Some scholars took glacier dynamics into account when they studied the Greenland region, and compared the differences of glacier sheet quality monitoring with GRACE through the climate model and ice sheet model simulation. Alexander et al. [22] found that some areas showed significant differences in peak values times in the annual cycle of mass change on sub-ice-sheet-wide scales, by GRACE with the simulation model; Schlegel et al. [23] found that hydrological and ice–ocean interaction processes should be considered in the model with relatively high temporal resolution (from month to season) to achieve an accurate estimation of Greenland's mass balance; Xu et al. [24] found that the approximate mass balance between GRACE and input–output method (IOM) is consistent in most Greenland Ice Sheet regions, and the difference in the northwest may be due to underestimating the uncertainty of IOM solutions; Flowers [25] found that about 60% of Greenland's mass loss in 1991–2015 is attributed to surface mass balance, the net difference between melting and snowfall. The remaining 40% is attributable to dynamic mass loss or uncompensated ice flowing into the ocean.

However, some scholars did not consider glacier dynamics in the process of ice sheet mass analysis. They used different institutions and data sources to study the trend and acceleration of mass change over Greenland's entire ice sheet over different periods of time. For example, Velicogna et al. [26–28] earlier used GRACE data to study the mass balance of Greenland ice sheet, which concluded that the total melting rate of the Greenland ice sheet showed an increasing trend of $248 \pm 36$ km$^3$/a, and an acceleration of $-30 \pm 11$ Gt/yr$^2$; Ramilli et al. [29] estimated that the melting rate of the Greenland ice sheet was $-109 \pm 9$ Gt/a from 2002 to 2005; Slobbe et al. [30] used GRACE post-processing data to compare Greenland ice sheet mass changes from four different organizations. The results indicated that the different data sources caused different results; Baur et al. [31] explored the annual average Greenland ice sheet melting with a rate of $162 \pm 11$ km$^3$/a through GRACE RL04 data during 2002–2008; Joodaki et al. [32] found that the mass of the Greenland ice sheet melted at the rate of $-166 \pm 20$ Gt/a and the acceleration of melting was $-32 \pm 6$ Gt/a by GRACE RL04 data from 2002 to 2011; Lu Fei et al. [33] found that the melting speed and acceleration of the ice sheet were $-157.8 \pm 11.3$ Gt/a and $-17.7 \pm 4.5$ Gt/a$^2$, respectively, through GRACE RL05 data from 2003 to 2012. In addition, the melting rate significantly increased after 2010, from $-132.2$ Gt/a in 2003–2009 to $-252.5$ Gt/a in 2010–2012; Forsberg et al. [34] concluded that the mass change rate of the Greenland ice sheet was $265 \pm 25$ Gt/a, and the correlation coefficient was 0.72 with the global mean sea level



change. In addition, some scholars studied ice sheet mass trends in different regions of Greenland. Chen et al. [35,36] reported that the mass change rate of Greenland ice sheet was $-239 \pm 23$ km$^3$/yr from April 2002 to November 2005. In addition, its decrease rate increased in the northwest and tended to be balanced in the southeast between 2007 and 2009; Wouter et al. [37] found that the overall mass change rate of the Greenland ice sheet was $-179 \pm 25$ km3/yr on a smaller scale from February 2003 to January 2008 and the greatest mass loss occurred in the southeastern coast (279 Gt) and the northwest coast (328 Gt) in the summer of 2005 and 2007; Zhu Chuandong et al. [38] found that the annual total melting amount of the Greenland ice sheet was $188 \pm 10$ km$^3$/a during 2002–2011, and the melting area mainly concentrated in the southeast and northwest of the ice sheet; Shamshiri et al. [39] used GRACE RL05 data to conclude that the peak loss of ice mass was $-15$ cm/yr in the southeast and northwest of Greenland, and loss acceleration was $-2.5$ cm/yr$^2$ in the southwest during 2003–2014.

Previous studies not only covered a relatively short period, but also mainly focused on the trend of time series, so there were few studies on the significant distribution of spatial trends and the driving factors of ice sheet mass loss. In this paper, we used longer and the latest time series GRACE RL06 data from January 2003 to December 2015 to invert the mass balance of the Greenland ice sheet, and analyze the change trend of ice sheet mass in Greenland. Then, the spatial–temporal variations of the Greenland ice sheet mass were analyzed by using the empirical orthogonal function (EOF) decomposition method. Finally, the common power, relative phase, and correlation of the time series of the Greenland ice sheet mass change and land surface temperature were derived by using continuous wavelet transform (CWT) and cross wavelet transform (XWT) in a time-frequency domain. The research results will help us to macroscopically understand the relationship between ice sheet mass change and land surface temperature in Greenland, which is of great significance for understanding global climate change.

## 2. Study Area and Data

### 2.1. Study Area

Greenland is the third largest country of North America and the largest non-continental island in the world [40] (see Figure 1). It lies between longitudes 11° and 74°W, and latitudes 59° and 83°N. In addition, Greenland is bordered by the Greenland Sea to the east, the Arctic Ocean to the north, Baffin Bay to the west, and the North Atlantic Ocean to the southeast. Meanwhile, Iceland is located in the southeast of Greenland in the Atlantic Ocean. Including other offshore minor islands, the total area of Greenland is 2,166,086 km$^2$ (836,330 sq mi). Among them, the Greenland ice sheet has a volume of about 2,850,000 km$^3$ (680,000 cu mi) and covers 1,755,637 km$^2$ (677,855 sq mi) (81%) [41].

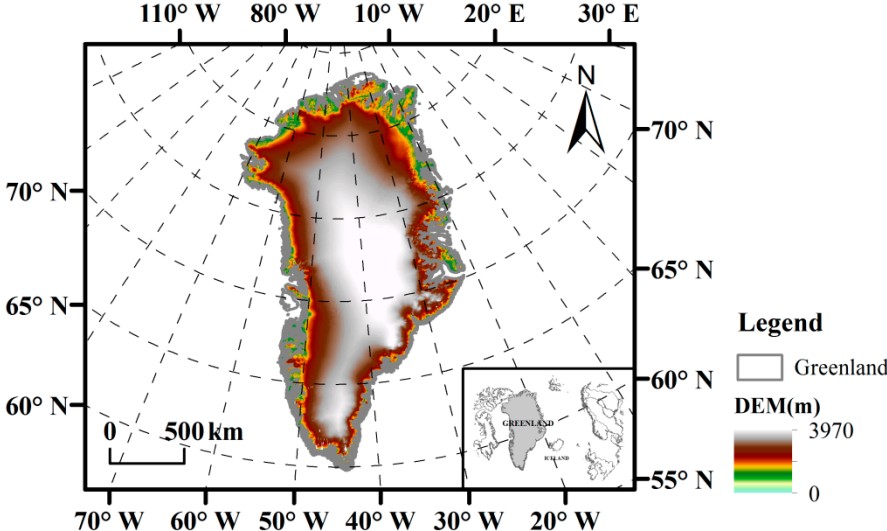

**Figure 1.** Map and topography of the Greenland in this study.

## *2.2. GRACE Data*

In this study, we use monthly GRACE data that are processed by the (CSR), University of Texas in Austin, to investigate Greenland ice sheet mass change. The CSR RL05 Level-2 data products are available for data spanning from April 2002 through July 2015, and it does not cover the remaining months of 2015. However, the latest release of GRACE Level-2 RL06 data (http://icgem.gfz-potsdam. de/grace/Level-2/CSR/RL06) covers the whole study period, and has some advantages in spatial resolution, accuracy, and periodic variation characteristics compared with RL05 [42,43]. So, we used CSR RL06 data up to a degree (l) and order (m) 60 in the form of spherical harmonics to study, which includes 142 monthly datasets from January 2003 to December 2015, and compensated for missing data with linear interpolation. Then, one can evaluate monthly local changes of the Earth's surface mass by using the time-varying gravity spherical harmonic coefficients provided by GRACE [4], with a time resolution of one month, and spatial resolution of $1° \times 1°$:

$$\Delta\sigma = \frac{a\rho_{ave}}{3} \sum_{l=0}^{l_{max}} \sum_{m=0}^{l} \frac{2l+1}{1+k_l} (\Delta C_{lm} \cdot \cos(m\lambda) + \Delta S_{lm} \cdot \sin(m\lambda)) \widetilde{P}_{lm}(\sin(\theta)) \tag{1}$$

where $\Delta\sigma$ is terrestrial water storage change (surface density variation), $a$, $\rho_{ave}$, $\theta$, $\lambda$, $k_l$, $\Delta C_{lm}$, $\Delta S_{lm}$ are the equatorial radius of the Earth (6377 km), mean density of Earth (5517 kg/m$^3$), latitudes, longitudes, love numbers of degree l with $l_{max}$ = 60 [44,45], and coefficients changes of the normalized complex spherical harmonic (Stokes' coefficients), respectively. Meanwhile, $\widetilde{P}_{lm}(\sin(\theta))$ is the fully normalized Legendre function of degree l and order m.

In order to eliminate the influence of Earth's long period and average gravity field, the spherical harmonic coefficients changes are relative to the mean of the 142 monthly solutions. In data processing, the Degree 2 coefficients provided by SLR were used to replace C20 terms based on GRACE orbital solution [46], the degree 1 coefficients provided by Swenson et al. [47] were used to correct the variation terms of Earth's center of mass (i.e., S11, C11, and C10). Because GRACE observations cannot separate the effects of Glacial Isostatic Adjustment (GIA), the GIA model of Geruo et al. [48] was used to deduct its effects on GRACE observations.

In addition, due to the combined effects of satellite orbit error, ocean–atmosphere model error and the correlation error of the spherical harmonic coefficient of Earth's gravity field, there are obvious north–south "stripes" error and high frequency error [49] in the inversion of surface mass changes by using Formula (1). Considering the limitation of single filter, this paper used improved P4M6 de-correlation filter to remove north–south "stripes" errors [50], and used a Gaussian filter weight function with a 250 km radius to smooth the noise of the high-order spherical harmonic coefficient [51]. The calculation model is

$$\Delta\sigma = \frac{a\rho_{ave}}{3} \sum_{l=0}^{l_{max}} \sum_{m=0}^{l} \frac{2l+1}{1+k_l} W_l(\Delta C_{lm} \cdot \cos(m\lambda) + \Delta S_{lm} \cdot \sin(m\lambda)) \widetilde{P}_{lm}(\sin(\theta)) \tag{2}$$

$$W_l = -\frac{2l+1}{b} W_l + W_{l-1} \tag{3}$$

$$W_0 = \frac{1}{2\pi}, \ W_1 = \frac{1}{2\pi}\left(\frac{1+e^{-2b}}{1-e^{-2b}} - \frac{1}{b}\right) \tag{4}$$

$$b = \frac{\ln(2)}{1 - \cos(r/a)} \tag{5}$$

where $W_l$ is a Gaussian-averaging kernel function related to order, and $r$ is the Gaussian filter smooth radius [4].

Finally, we note that the use of spherical harmonic coefficients of a finite order and spatial filtering leads to 'leakage' errors in data processing. However, scale factor 1.95, which was calculated by Velicogna and Wahr [22] could be used to minimize the combined signal leakage and measurement

error. In addition, in order to evaluate ice sheet mass changes, surface mass anomalies can be calculated by integrating over the area of specific, since $\frac{\Delta\sigma}{\rho_w}$ is interpreted as the equivalent water height (i.e., in mm), in that $\rho_w$ is the density of water ($10 \times 10^3$ kg/m$^3$). The ice sheet mass variation $\Delta m$ of Greenland can thus be given as [4]

$$\Delta m = \int \Delta\sigma(\theta,\lambda)\tau(\theta,\lambda)dA \tag{6}$$

$$dA = (\frac{\pi}{180})^2 \cos(\theta)d\theta d\lambda \tag{7}$$

$$\tau(\theta,\lambda) = \begin{cases} 1, & inside \quad the \quad region \\ 0, & outside \quad the \quad region \end{cases} \tag{8}$$

where $\theta$, $\lambda$, and $dA$ are the latitudes, longitudes, and surface element, respectively. In addition, $\tau(\theta,\lambda)$ is a region smoothing kernel function, which can be used to accurately evaluate the regional variation of ice sheet mass in Greenland.

### 2.3. Land Surface Temperature data

GHCN CAMS (Fan et al., 2008) [52] data were downloaded from the website (http://www.esrl. noaa.gov/psd/data/gridded/tables/.temperature.html) of the National Oceanic and Atmospheric Administration (NOAA), which is a high resolution ($0.5° \times 0.5°$) analyzed global land surface temperature (LST) from 1948 to near present. In this study, GHCN CAMS data were used to study the relationship between ice sheet mass change and land surface temperature in Greenland.

## 3. Methods

### 3.1. Theil–Sen Median Trend Analysis

Theil–Sen median trend analysis is a robust non parametric statistical trend calculation method proposed by Hirsch and Slack [53] and Kumar and Saharia [54], which can reduce the influence of abnormal values [55]. For example, this method can calculate the slope of equivalent ice sheet mass from GRACE and temperature changes to analyze trends. The formula is

$$S_V = \quad Median(\frac{V_j - V_i}{j - i}) \quad (2003 \leq i < j \leq 2015) \tag{9}$$

where $S_V$, $i$, and $j$ denote the slopes, and time points, respectively. In addition, $V_j$ and $V_i$ are values at the points of $j$ and $i$, respectively. The negative value means a decreasing trend, and the positive value means an increasing trend [53].

### 3.2. Mann–Kendall (MK) Trend Test

The nonparametric M–K test is widely used for trend analysis and significance test of hydrometeorological elements such as rainfall, temperature, and runoff. It has a simple structure, convenient calculation and is not disturbed by a few abnormal values [56]. Statistic S could be obtained based on Equations (1) and (2) [57]

$$S = \sum_{i=1}^{n-1} \sum_{j-i+1}^{n} sgn(x_j - x_i) \tag{10}$$

$$sign(x_j - x_i) = \begin{cases} 1, & (x_j - x_i > 0) \\ 0, & (x_j - x_i = 0) \\ -1, & (x_j - x_i < 0) \end{cases} \tag{11}$$

where $n$ is the number of data sets, $x_j$ and $x_i$ are values at $j$th and $i$th points, and sign is a sign function. Then, the M–K statistic, $Z$, is calculated as

$$Var(S) = \frac{n(n-1)(2n+5)}{18} \tag{12}$$

$$Z = \begin{cases} (S-1)/\sqrt{Var(S)} & (S > 0) \\ 0 & (S = 0) \\ (S+1)/\sqrt{Var(S)} & (S < 0) \end{cases} \tag{13}$$

Given a significant level of $\alpha$, when $|Z| > Z_{1-\alpha/2}$, there is a significant change in the study sequence at the level of $\alpha$. The confidence limit was $\alpha = 0.05$ in this study, which means $|Z| > Z_{1-\alpha/2} = 1.96$.

### 3.3. Rotated EOF Method

Empirical orthogonal function analysis is also called principal component analysis or feature vector analysis. It can be used to analyze the variance contribution of different components for extracting the spatial–temporal feature of TWS [58]. The details are

$$Y = (y_1, y_2, \cdots, y_p) = \begin{bmatrix} y_{11} & \cdots & y_{1p} \\ \vdots & \ddots & \vdots \\ y_{n1} & \cdots & y_{np} \end{bmatrix} \tag{14}$$

where $n$ and $p$ denote the number of space point, the time period, respectively. In addition, the covariance matrix $C$ of time series is obtained and orthogonally decomposed as

$$C = \frac{1}{p} YY^{\mathrm{T}} \tag{15}$$

$$C = E\Lambda E^{\mathrm{T}} = (e_1, e_2, \cdots, e_p) = \begin{bmatrix} \lambda_1 & \cdots & 0 \\ \vdots & \ddots & \vdots \\ 0 & \cdots & \lambda_n \end{bmatrix} \begin{bmatrix} e_1^{\mathrm{T}} \\ e_2^{\mathrm{T}} \\ \vdots \\ e_n^{\mathrm{T}} \end{bmatrix} \tag{16}$$

where $\lambda_1 > \lambda_2 > \cdots > \lambda_p$, the orthogonal decomposition of space domain can be obtained by calculating the feature vector of $YY^{\mathrm{T}}$, and the *EOF* value can then be obtained by normalizing the decomposition. In addition, the principal component $Z$ can be calculated in the time series.

$$EOF = E * diag(\frac{1}{\sqrt{\sum E^2}}) \tag{17}$$

$$Z = E^{\mathrm{T}} Y \tag{18}$$

### 3.4. Continuous and Cross Wavelet Transform

Firstly, we briefly describe the CWT. The CWT definition of time series ($X_n$, $n = 1, 2, \cdots, N$) is [59]

$$W_n^X(s) = \sqrt{\delta t/s} \sum_{n'=1}^{N} X_{n'} \varphi_0 \left[ (n' - n)\frac{\delta t}{s} \right] \tag{19}$$

where $W_n^X(s)$ is the wavelet coefficient, $s$ is the wavelet scale, $\varphi_0$ and $\delta t$ are the mother wavelet function and the time scale, respectively, and $\sqrt{\delta t/s}$ and $n'$ represent the normalization factor and the reversed time, respectively. The idea of wavelet transform is that the wavelet as a band-pass filter is applied to time series where the characteristic period of the filter is linearly related to the wavelet scale.

XWT is a new signal analysis technology that combines cross-spectrum analysis with wavelet transform, and studies the relationship between two time series from multi-time scales in the time–frequency domain. Temperature is a well-known trigger for ice sheet melting. Hence, XWT was used to analyze Greenland ice sheet mass changes and temperature time series, which can reveal the regions in which two nonlinear time series with a consistent phase relationship with high common power. The cross wavelet transform of two time series $X_n$ and $Y_n$ is given as [60–62]

$$W^{XY} = W^X W^{Y*} \tag{20}$$

where $W^{Y*}$ denotes a complex conjugation of $W^Y$. The cross wavelet power spectrum is defined as $|W^{XY}|$, and the complex argument of $W^{XY}$ can be seen as the local relative phase between $X_n$ and $Y_n$ in time–frequency domain. While the larger the $|W^{XY}|$ value, indicating that $X_n$ and $Y_n$ have a common high-energy region, which significantly correlates them with each other. More details about Cross Wavelet Power Spectrum test and phase difference calculation are given by Grinsted et al. [59] and are not repeated here.

In analysis, we use the circular mean of the phase and XWT-based semblance to quantify the phase relationship and correlation between two time series. The circular mean of a set of angles ($\alpha_i$, $i = 1, 2, \ldots, N$) is defined as

$$a_m = a \tan 2(X,Y) = a \tan 2(\frac{1}{n}\sum_{i=1}^{n}\cos(\alpha_i), \frac{1}{n}\sum_{i=1}^{n}\sin(\alpha_i)) \tag{21}$$

We can use circular standard deviation $s = \sqrt{-2\ln(R/n)}$ ($R = \sqrt{X^2 + Y^2}$) to qualify the scatter of angles around the mean. The XWT-based semblance is defined as [63]

$$\rho = \cos(\alpha_i) \tag{22}$$

The range of the $\rho$ value is $-1$ to $0$ to $1$, which is expressed as negative correlation, irrelevant correlation and positive correlation, respectively. Among them, the closer the value of $\rho$ is to 1, the greater the correlation. In this analysis, we compare the two time series by using XWT-based semblance.

## 4. Results and Discussion

### 4.1. Time Variation Analysis

#### 4.1.1. Time Series and Change Trends

For Greenland, we estimated the total ice sheet mass change for each month. As it is shown in Figure 2, the time series variation of monthly Greenland ice sheet mass was estimated in Gt. With the least squares estimate, we fit a linear trend in a mass loss of $-268 \pm 12$ Gt/yr and a quadratic trend in an acceleration of $-11 \pm 2$ Gt/yr$^2$ in 2003–2015, as was done in most previous studies. We use a quadratic form to obtain a trend of $-195 \pm 21$ Gt/yr for the Greenland ice sheet between 2003 and 2015. The uncertainties of the data results ($\pm 12$, $\pm 2$, and $\pm 21$) include contributions from the gravity field error, signal leakage effects, truncation error, GIA correction, and the statistical uncertainty of the fit. Mass loss increases with time in a relatively consistent pattern from 2003 to 2012, and a sudden decline in the time series could be observed in 2012. This was followed by a relatively stable mass loss in the next two years after 2012, and then continued in a decreasing trend. The above results are basically consistent with those obtained by researchers using GRACE data in recent years [27,28,33,36,39,64]. However, the time series of the results of this study are longer than those of previous studies, especially the trend of change in 2015 relative to 2014, and their different values are due to different data sources, time series, and post-processing methods.

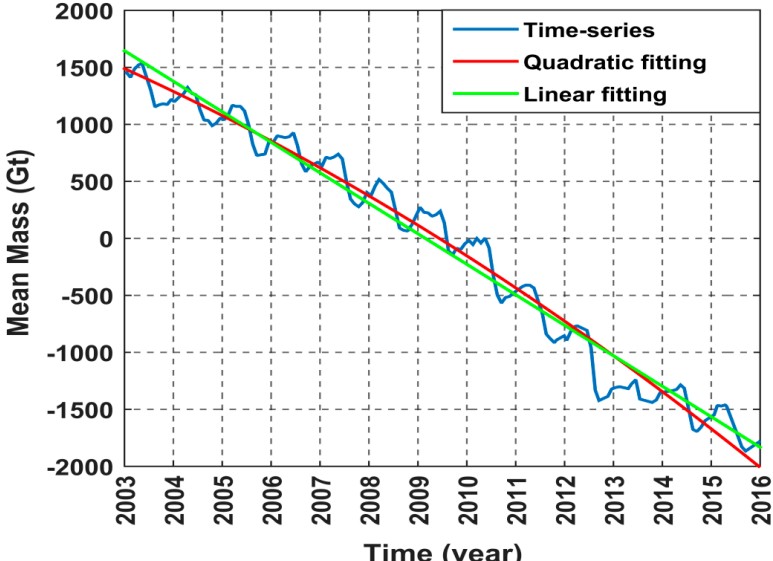

**Figure 2.** Monthly mass change of Greenland from January 2003 to December 2015 estimated based on the GRACE solution of the CSR processing center. The blue line, green line and red curve represent the mass change of the time series, and the best fitting of linear and quadratic trend, respectively.

Then, we used the adjusted R-squared ($R^2_{Adj}$) of the data fit to investigate which of the quadratic or linear models is more suitable for time series fitting. More details about $R^2_{Adj}$ were computed by Johnson et al. [65] and are not repeated here. Finally, we found that the $R^2_{Adj}$ for the linear model is 0.9713, which is smaller than the 0.9763 value of the quadratic one. The result agrees with those of other researchers [27,32,64].

### 4.1.2. Monthly Mean and Seasonal Change

In order to investigate ice sheet mass of intra-annual variability, we calculate the monthly and seasonal changes of ice sheet mass in Greenland, respectively, which are shown in Figure 3. Seasonal changes in the ice sheet mass were averaged from the ice sheet mass in the corresponding months.

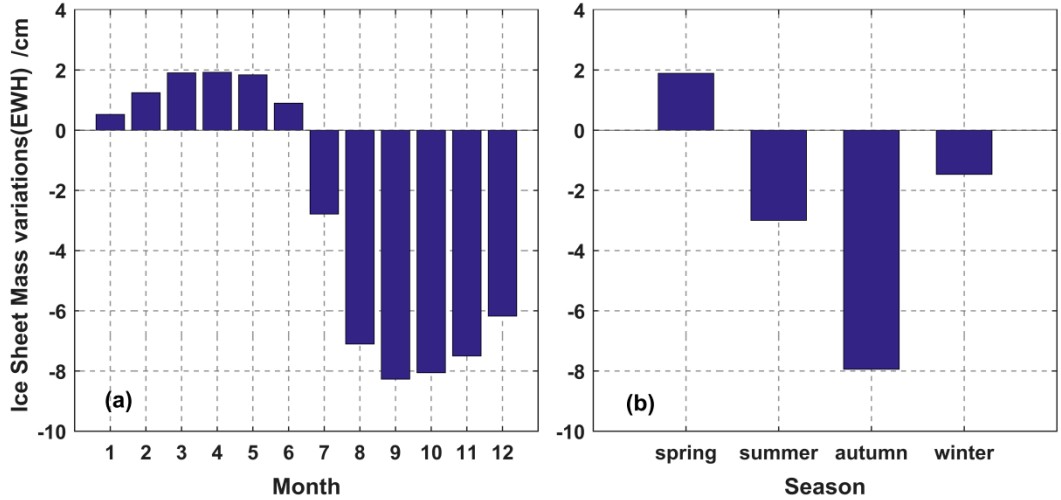

**Figure 3.** (**a**) and (**b**) represented monthly and seasonal changes in the mass of the Greenland ice sheet from January 2003 to December 2015, respectively.

From Figure 3a, it can be found that positive ice sheet mass change values from GRACE primarily occurred in the first half of each calendar year, while negative ones existed in the second half of each

calendar year. The change of ice sheet mass in the whole study area slowly increases between October and April, but decreases between May and September, indicating that the ice sheet mass starts to decrease from May and increase from October. This conclusion is consistent with other scholars, but there are still some differences with regional changes, mainly due to different factors in different regions [22,23,32].

From Figure 3b, it can be found that the average change of ice sheet mass in the whole study area is positive in spring and negative in the rest of the year. Meanwhile, we can also see that the mass of ice sheet decreases from spring to autumn, and then increases from autumn to winter, which is consistent with monthly ice sheet mass changes.

*4.2. Spatial Change Analysis*

To further study the change trend of ice sheet mass in Greenland from 2003 to 2015, this paper used Theil–Sen median trend analysis and the M–K test to analyze the change trend and significant spatial distribution characteristics of the ice sheet mass (GRACE), and then obtain pixel scale change trends in the whole study area.

In Greenland, we distinguish Greenland into five regions according to the division principle of van den Broeke et al. [66]: (1) southeast (SE), which has a high accumulation, a large number of glaciers at its outlet, a fast speed and a small export; (2) northwest (NW), with highly accumulated and rapidly moving glaciers; (3) southwest (SW), where most glaciers are terrestrial terminals, with the largest melting area; (4) north (N), with a small number of low accumulation, large, and slow-moving glaciers; and (5) northeast (NE), which is similar to N, but glaciers flow through alpine mountains.

Figure 4 shows the spatial distribution of the inter-annual mass loss trend and significance of the Greenland ice sheet from 2003 to 2015, which showed an increasing trend as a whole. The mass loss areas of the Greenland ice sheet are mainly concentrated in the southeastern, southwestern, and northwestern regions, which passed the M–K test of 95% confidence interval. Especially in southeastern Greenland, the trend of mass loss is the largest, with a significant increase rate greater than 27 cm/yr (equivalent water height). In addition, although mass loss rate is less than 3 cm/yr in the northeastern region, it still passes the M–K test of 95% confidence interval, indicating a significant change trend. The above results are consistent with previous studies on regional trends of ice sheet mass loss [28,33].

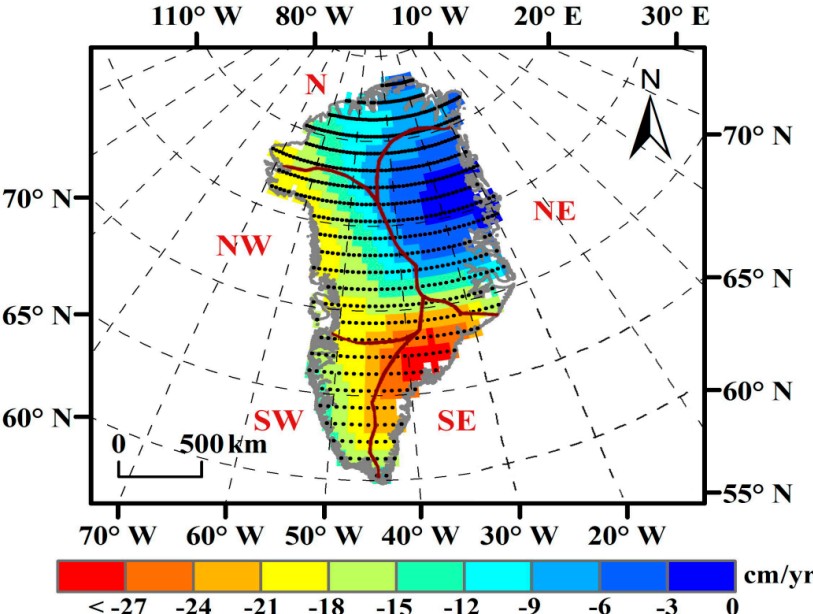

**Figure 4.** Spatial distribution of annual change trend of GRACE ice sheet mass in Greenland from 2003 to 2015. Note: Lattice point representation has passed the M–K test of 95% confidence interval.

### 4.3. EOF Analysis

In order to analyze the spatial distribution characteristics of the mass change of the Greenland ice sheet, the data of GRACE ice sheet mass change were decomposed by EOF from 2003 to 2015. The first second PCA modes account for 98.6% and 0.6% of the variance, respectively, whose accumulative variance is 99.2%; we show the spatial distribution and time-varying characteristics in Figure 5.

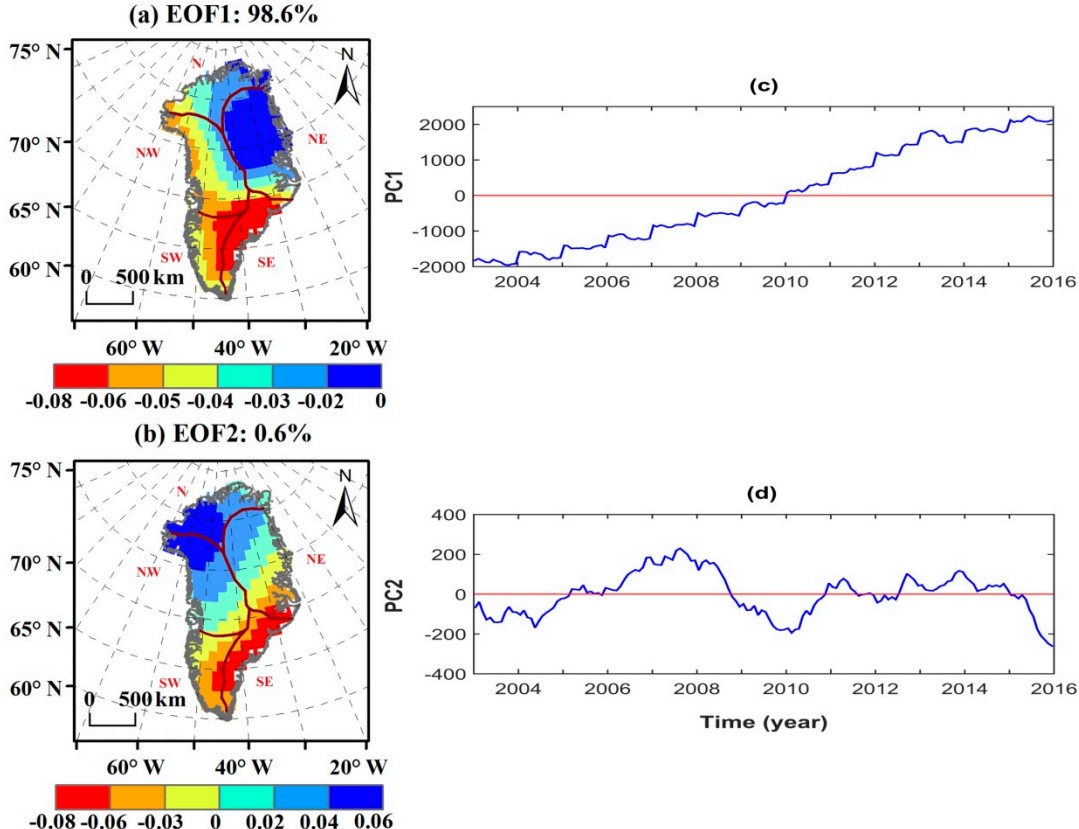

**Figure 5.** Spatial distribution (**a**,**b**) and time coefficient (**c**,**d**) change of the first and second EOF modes. Note: The time coefficients and EOF variables in the figure indicate relative size and are dimensionless.

Figure 5a shows that the overall trend of temporal and spatial variations in Greenland ice sheet mass. EOF1 increases around the southeastern region of Greenland with strong negative values, and the whole region shows negative values, with the northeastern region being the smallest negative value. In terms of time coefficients (i.e., Figure 5c), there is an obvious upward trend from 2003 to 2015, which indicates that the change of ice mass in southeastern Greenland has had a downward trend in the past 13 years. Among them, time coefficients change was obviously weakened in 2010 and 2012, which indicates that the decreasing trend of the ice sheet mass was obvious in 2010 and 2012. At the same time, by comparing ice sheet mass change curve in Figure 2 and the ice sheet mass spatial change significant distribution map in Figure 4, we can also find the corresponding rules.

Figure 5b is the second eigenvector and time coefficient (variance contribution value, 0.6%) derived from the EOF decomposition of GRACE ice sheet mass change in Greenland. As shown in Figure 5b, the southeastern region of Greenland is strongly negative, while the northwestern region is strongly positive, and the other regions are not obvious. In terms of time coefficients (i.e., Figure 5d), there was a significant upward trend from the first half of 2004 to the second half of 2007, and from the first half of 2010 to the first half of 2011, which indicates that the mass loss of the ice sheet in southeastern Greenland was increasing while that in northwest Greenland was decreasing. This result is consistent with that of Lu et al. [33], but the time coefficients from the second half of 2007 to the first half of 2010 and from 2014 to 2015, significantly decreased, which indicates that the loss of ice sheet

mass in southeastern Greenland was decreasing while that in northwestern Greenland was increasing. In addition, the change of ice sheet mass in other regions is relatively stable.

### 4.4. Relationship between Temperature and Ice Sheet Mass

#### 4.4.1. Spatiotemporal Contrastive Analysis

In a time series, in order to eliminate the influence of ice sheet mass and land surface temperature changes within a year, the original time series data were averaged from 13 points, and the Greenland ice sheet mass (GRACE) and land surface temperature change maps were obtained for the study period. Figure 6 shows that the change of GRACE ice sheet mass is obviously opposite to the change of land surface temperature, which reflects the correlation between them to a certain extent. Although there are some differences between them, the main peak–valley characteristics of ice sheet mass changes can be well identified by land surface temperature changes, and the peak–valley value is 1–2 months earlier than the ice sheet mass changes. Overall, the land surface temperature changes show a downward trend (a linear change equation of y= −0.052x + 93) during the study period, which can explain the decline of ice sheet mass to a certain extent.

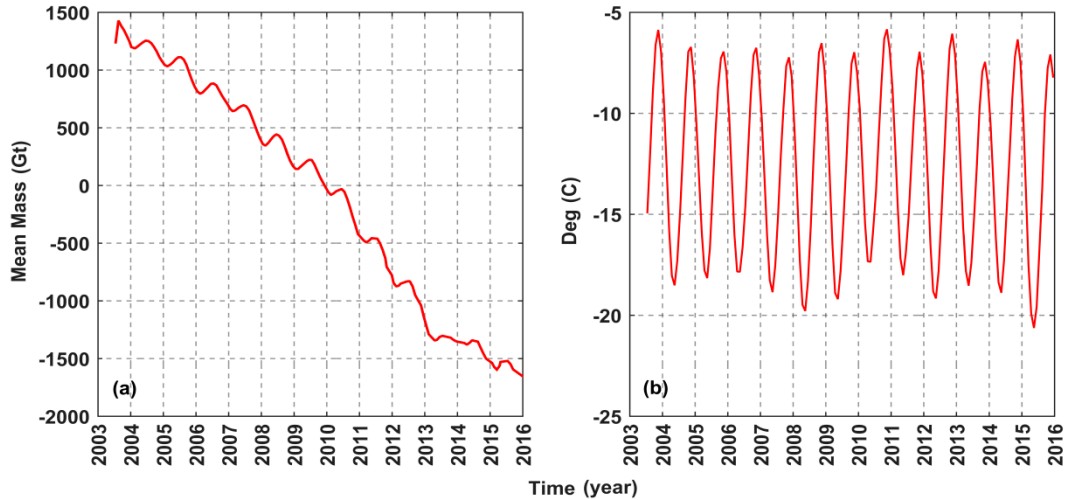

**Figure 6.** Annual changes of GRACE ice sheet mass and land surface temperature in Greenland from 2003 to 2015.

Compared with Figures 4 and 7, it can be found that the land surface temperature changes from the southeast to the southwest of Greenland show a significant downward trend, while ice sheet mass loss is significant. In the northwestern region of Greenland, land surface temperature rise is particularly obvious, while the trend of ice sheet mass loss is not the most significant change, which is mainly caused by glacial discharge, temperature, and snow cover [66]. Land surface temperature changes in the northeastern part of Greenland are not significantly decreasing, and the loss change of the ice sheet mass also showed a small but significant reduction. In summary, the change of ice sheet mass and land surface temperature in Greenland are correlated with changes in spatially significant distribution.

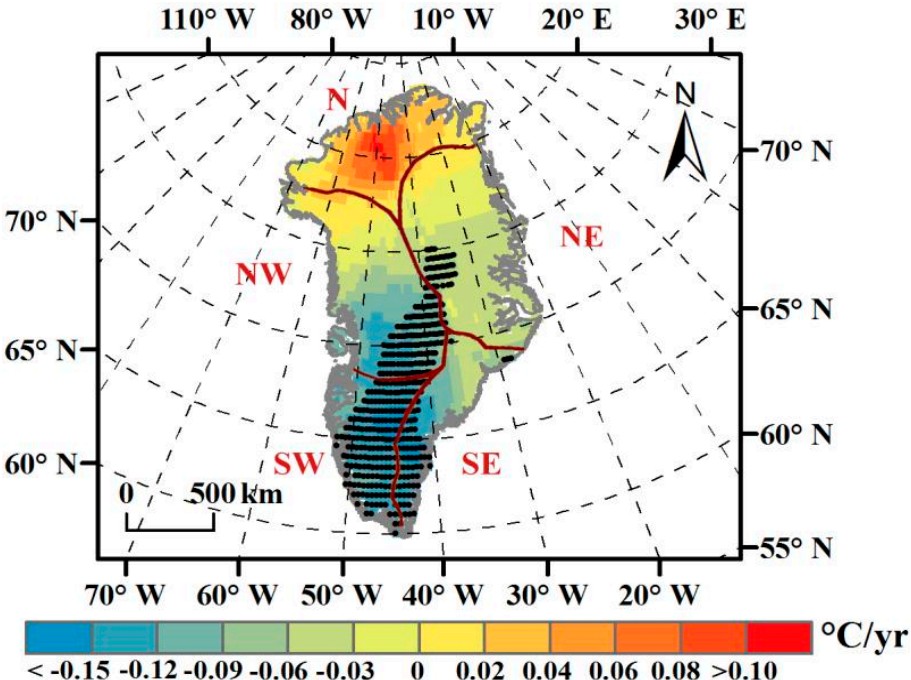

**Figure 7.** Spatial distribution of annual change trend of GHCN CAMS land surface temperature in Greenland from 2003 to 2015. Note: Lattice point representation passed the M–K test of 95% confidence interval.

### 4.4.2. Wavelet Transform Analysis

In order to study the relationship, continuous wavelet transform (CWT) and XWT were used to detect a common power and common relative phase between the time series of ice sheet mass changes and the time series of land surface temperature in Greenland. These wavelet tools can be used to detect local intermittent periodicity [67]. The National Oceanography Center provided the MATLAB code we used [68]. The thick, black contour line indicates that the 95% confidence test was passed, and the thin, black line represents the influence cone (COI), which defines the area not affected by the edge effect. As shown in Figure 8, the energy distribution of the wavelet power spectrum of land surface temperature and ice sheet mass changes is generally consistent, and peak value is mainly concentrated in 10–14 months. In the whole study period, we could see a clear annual cycle, the 12-month spectral energy passed the 95% confidence test, and the shape of the significant band of the wavelet power spectrum in this frequency band is very similar. Results show that the change of land surface temperature and ice sheet mass has a significant 12-month oscillation period.

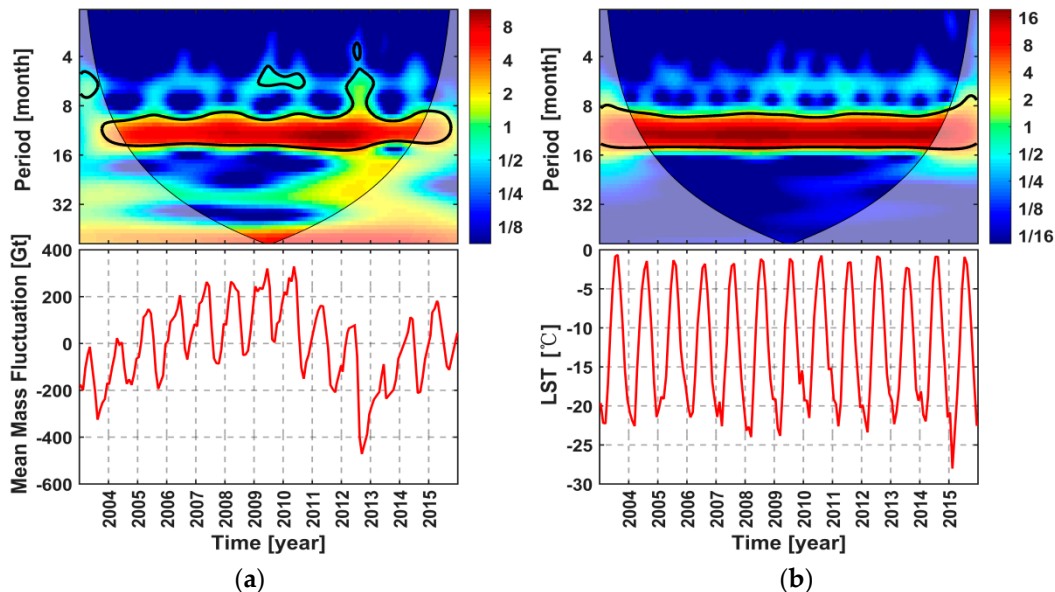

**Figure 8.** Continuous wavelet transform with average mass and land surface temperature represented by (**a**) and (**b**) from 2003 to 2015.

Similarly, the cross-wavelet power spectrum of land surface temperature and ice sheet mass changes in Greenland could easily be calculated (Figure 9). The arrow direction reflects the relative phase relationship [36] in Figure 9 (with anti-phase pointing left, in-phase pointing right, ice sheet mass changes leading land surface temperature changes by 90°, pointing straight down, and land surface temperature changes leading ice sheet mass changes by 90°, pointing straight up). From Figure 9, the red high-energy region shows that the common power of the two time series was higher for 10–14 months. Since the annual period of 12 month was dominant in the whole observation period, this study mainly focused on the relative phase angles in the 12 month period band, and its average phase angle is $-110° \pm 6°$ (6°denotes circular standard deviation).

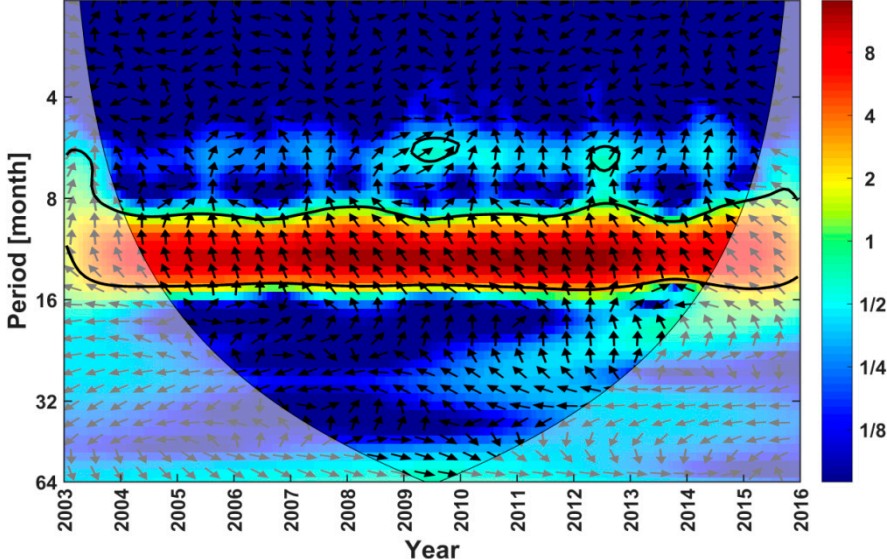

**Figure 9.** Cross wavelet transform of ice sheet mass and land surface temperature fluctuations. Arrows indicate relative phase relations, where straight-up arrows represent ice sheet changes, and land surface temperature shows an anti-phase relationship.

Figure 10 describes the XWT-based semblance curve at the period that is closest to one cyclic period year (cpy) outside COI to better show the correlation. The mean XWT-based semblance is −0.34, indicating that the ice sheet mass and land surface temperature changes have an anti-phase relationship in the whole observation period of a significant common power. This correlation is not high. On the one hand, this is because land surface temperature data are surface (3 m) temperature, not ice sheet temperature, so accuracy is relatively low. On the other hand, ice sheet mass change is a combination of increased snow accumulation, increased coastal glacier discharge, and increased surface temperature.

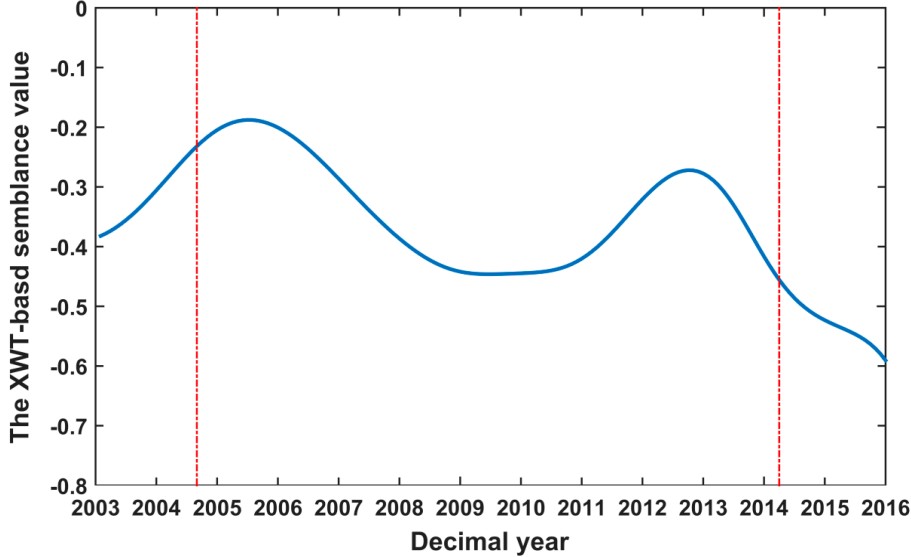

**Figure 10.** XWT-based semblance value is the closest to one cpy outside the COI. Red dashed lines represent the COI, which limits the region not affected by edge effects.

## 5. Conclusions

In this study, the GRACE and GHCN CAMS data were used to investigate the spatiotemporal distribution of ice sheet mass changes and their relationship with land surface temperature in Greenland. GRACE data were used to calculate the mass changes of the Greenland ice sheet from January 2003 to December 2015. The major conclusions of this study are as follows:

From 2003 to 2015, the total mass of the ice sheet decreased steadily at a speed of $-195 \pm 21$ Gt/yr and an acceleration of $-11 \pm 2$ Gt/yr$^2$, which was relatively stable for the two years after 2012, and then continues to show a downward trend. However, in terms of monthly and seasonal variations, the change of ice sheet mass in the whole study area slowly increases between October and April, but decreases between May and September.

According to the inter-annual and spatial variation trend of ice sheet mass and land surface temperature in Greenland, it could be found that ice sheet quality in different regions of Greenland has a significant change trend. For example, Greenland ice sheet mass loss areas are mainly concentrate in the southeastern, southwestern, and northwestern regions, and the southeastern region mass losses at a maximum rate of more than 27 cm/yr (equivalent water height), while northeastern region losses happen at a minimum rate of less than 3 cm/yr. In addition, for EOF results, the variance contribution of the first two eigenvectors is 98.6% and 0.6% respectively, totaling 99.2%. Through the spatial distribution and time coefficients of the first two models, it can be seen that overall ice mass change in Greenland significantly declined in 2010 and 2012. Meanwhile, the ice sheet mass changes in the southeastern and northwestern regions of Greenland showed different significant changes in different periods from 2003 to 2015, while the other regions showed relatively stable changes.

In the time–frequency domain, continuous and cross-wavelet transform were used to study the seasonal relationship between the two time series. We found that land surface temperature

and ice sheet mass changes had a significant oscillation periodicity in the 12-month period, and the mean XWT-based semblance value was −0.34, which indicates that ice sheet mass and land surface temperature changes had anti-phase relationships in the whole observation period of a significant common power. In addition, the XWT-based semblance values had the largest relative change in 2005 and 2012, and the smallest relative change in 2009 and 2010, which indicates that the influence of land surface temperature on ice sheet mass significantly varies in different years. However, there are some limitations in the analysis of the relationship between land surface temperature and ice sheet mass changes. It is necessary to take glacier dynamics into account in order to achieve more accurate analysis results in the future.

**Author Contributions:** Y.B. and J.Y. designed the study and performed the experiments; Y.B. wrote the draft of the manuscript; J.Y. and W.G. supervised the research and revised the manuscript; the other authors are responsible for revision.

**Funding:** This work was financially supported by National Key R&D Program of China (2018YFC1508603).

**Acknowledgments:** We are very grateful to Center for Space Research (CSR) for providing RL06 data, National Oceanic and Atmospheric Administration (NOAA) for providing GHCN CAMS data. The authors thank Wei Feng, Vagner G. Ferreira, and Yanping Cao for their help to provide guidance, and would like to thank the editor and three anonymous reviewers for their time and constructive criticism.

**Conflicts of Interest:** The authors declare no conflicts of interest.

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
