# Peer review of "Analysis of the Spatiotemporal Changes of Ice Sheet Mass and Driving Factors in Greenland"

_remotesensing, doi:10.3390/rs11070862_

Round 1
Reviewer 1 Report
I recall reviewing the first version of this manuscript, and find that this resubmitted version is greatly improved. I do still see a need for further checking for english style, appropriate phrasing, and other minor text-related issues. The methods are well-described, and the results appear to be well-discussed. I would recommend publishing this manuscript with minor revisions.
Line 38: Freshwater ice or total ice? (including sea ice)
Line 39: Not sure "more serious" is the right express here... try "more pronounced"
line 50 - line 109: another major consequence of the large input of freshwater from the Greenland ice cap is the potential to change / alter / slow down the thermohaline circulation in the Atlantic Ocean... you may wish to touch on this a bit in your introduction / lit review.
Line 113: "determined" ... might consider "processed" instead?
Line 123: "Which's" is not proper grammar. Try "with a temporal resolution of 1 month..."
Line 154: "can be get" improper english. Try "can be retrieved" or "can be calculated"
Line 166: 0.5 x 0.5 missing units km? miles? degrees?
Line 221: "The Temperature"... should be "Air Temperature" or simply "Temperature"
Line 248: This sentence reads really odd.... has obvious seasonal changes? Are we talking a about one or many time series?
Line 306: Figure 3 shows the spatial distribution... not "is"
Line 322: ...and ARE dimensionless...
Author Response
Dear Reviewer:
We would like to express our appreciation to you for your thoughtful comments and constructive suggestions, which helped to enhance the quality of this manuscript, as well as the important guiding significance to our researches. According to your comments, we have revised our paper carefully and we believe that your comments have been addressed. In the following, we give an item-by-item response to your comments.
Point 1: The article is still needed for further checking for English style, appropriate phrasing, and other minor text-related issues.
Response 1: Thanks for your constructive and positive comments. In response to this problem, we have found a more authoritative polishing agency (Remote Sensing English editing) to revise the article.
English polishing proof picture:
Point 2: line 50 - line 109: another major consequence of the large input of freshwater from the Greenland ice cap is the potential to change / alter / slow down the thermohaline circulation in the Atlantic Ocean
Response 2: Thank you for your suggestion. In response to this problem, we have added a description of it in the introduction.
See line 56 – line 63: Among them, in terms of glacier melting, it is helpful to understand the impact on thermohaline circulation in the Atlantic Ocean. Murray found that the large freshwater input from the Greenland ice sheet may weaken or destroy the "thermo-salt" cycle of marine saltwater, thereby seriously altering the climate of the Northern Hemisphere[20]; Yang et al estimated new heat and salt flows from the North Atlantic to the Labrador Sea using updated GRACE satellite data, and suggested that the changes in Labrador Sea Water Density (LSW) might be related to the weakening of the Atlantic Meridional Overturning Circulation (AMOC) [21]. In addition, it also plays an important role in the study of the mass change of the Greenland ice sheet.
Special thanks to you for your good comments.

Reviewer 2 Report
Dear Authors:
Comments about the responses to the review of the
manuscript "Analysis of the spatiotemporal changes of ice sheet mass and
driving factors in Greenland" by Yankai Bian et al.
For my
part, much of my observations have been corrected, the references are
correctly cited, the figures are improved, the errors in the text have
been fixed.... also some sections of the manuscript has been improved.
I'm still missing a location map (as first figure, despite the figure 3
can be used as a location map). Despite this last point, I recommend the
publication of the manuscript.
Just a note:
Line
48: change "In March 2002, the Gravity Recovery and Climate Experiment
Satellite (GRACE)" by "In March 2002, the Gravity Recovery and Climate
Experiment (GRACE) Satellite" or by "In March 2002, the GRACE Satellite"
(because you've already defined GRACE)
Author Response
Dear Reviewer:
We would like to express our appreciation to you for your thoughtful comments and constructive suggestions, which helped to enhance the quality of this manuscript, as well as the important guiding significance to our researches. According to your comments, we have revised our paper carefully and we believe that your comments have been addressed. In the following, we give an item-by-item response to your comments.
Point 1: The article is still missing a location map (as first figure, despite the figure 3 can be used as a location map).
Response 1: Thank you for your suggestion. In view of this problem, we have added a regional distribution map of Greenland, and described it with relevant information. At the same time, I have made some modifications to Figure 4.
See line 119 - line 125:
Greenland is the third largest country of North America and the largest non-continental island in the world [40]. It lies between longitudes 11° and 74°W, and latitudes 59° and 83°N. In addition, Greenland is bordered by the Greenland Sea to the east, the Arctic Ocean to the north, Baffin Bay to the west, and the North Atlantic Ocean to the southeast. Meanwhile, Iceland is located in the southeast of Greenland in the Atlantic Ocean. In the case of other offshore minor islands, the total area of Greenland is 2,166,086 km2 (836,330 sq mi). Among them, the Greenland ice sheet has a volume of about 2,850,000 km3 (680,000 cu mi) and covers 1,755,637 km2 (677,855 sq mi)- (81%) [41].
Figure 1 and 4:
Figure 1. Map and topography of the Greenland in this study.
Figure 4. Spatial distribution of annual change trend of GRACE ice sheet mass in Greenland from 2003 to 2015.
Note: Lattice point representation has passed the M-K test of 95% confidence interval.
Point 2: change "In March 2002, the Gravity Recovery and Climate Experiment Satellite (GRACE)" by "In March 2002, the Gravity Recovery and Climate Experiment (GRACE) Satellite" or by "In March 2002, the GRACE Satellite"
Response2: Thanks for reminding us. We have amended this.
Point 3: The introduction and conclusion can be improved.
Response3: Thank you for your suggestion. For it, we have all made some amendments.
1. Introduction section:
See line 56 – line 63: Among them, in terms of glacier melting, it is helpful to understand the impact on thermohaline circulation in the Atlantic Ocean. Murray found that the large freshwater input from the Greenland ice sheet may weaken or destroy the "thermo-salt" cycle of marine saltwater, thereby seriously altering the climate of the Northern Hemisphere[20]; Yang et al estimated new heat and salt flows from the North Atlantic to the Labrador Sea using updated GRACE satellite data, and suggested that the changes in Labrador Sea Water Density (LSW) might be related to the weakening of the Atlantic Meridional Overturning Circulation (AMOC) [21]. In addition, it also plays an important role in the study of the mass change of the Greenland ice sheet.
2. Conclusions section:
See line 420 – line 421 (Previous article): The seasonal change of ice sheet mass and land surface temperature is obvious, and the seasonal change of ice sheet mass lags behind that of land surface temperature. —(This part is unimportant and has been deleted)
See line 441 – line 443: According to the inter-annual and spatial variation trend of ice sheet mass and land surface temperature in Greenland, it could be found that ice sheet quality in different regions of Greenland has a significant change trend.
Special thanks to you for your good comments.
